# Effects of Inbreeding on Microbial Community Diversity of *Zea mays*

**DOI:** 10.3390/microorganisms11040879

**Published:** 2023-03-29

**Authors:** Corey R. Schultz, Matthew Johnson, Jason G. Wallace

**Affiliations:** 1Institute of Bioinformatics, University of Georgia, Athens, GA 30602, USA; crs68219@uga.edu; 2Plant Breeding, Genetics, and Genomics, University of Georgia, Athens, GA 30602, USA; 3Crop and Soil Science, University of Georgia, Athens, GA 30602, USA

**Keywords:** maize, microbiome, inbreeding, heterosis, genetic variation

## Abstract

Heterosis, also known as hybrid vigor, is the basis of modern maize production. The effect of heterosis on maize phenotypes has been studied for decades, but its effect on the maize-associated microbiome is much less characterized. To determine the effect of heterosis on the maize microbiome, we sequenced and compared the bacterial communities of inbred, open pollinated, and hybrid maize. Samples covered three tissue types (stalk, root, and rhizosphere) in two field experiments and one greenhouse experiment. Bacterial diversity was more affected by location and tissue type than genetic background for both within-sample (alpha) and between-sample (beta) diversity. PERMANOVA analysis similarly showed that tissue type and location had significant effects on the overall community structure, whereas the intraspecies genetic background and individual plant genotypes did not. Differential abundance analysis identified only 25 bacterial ASVs that significantly differed between inbred and hybrid maize. Predicted metagenome content was inferred with Picrust2, and it also showed a significantly larger effect of tissue and location than genetic background. Overall, these results indicate that the bacterial communities of inbred and hybrid maize are often more similar than they are different and that non-genetic effects are generally the largest influences on the maize microbiome.

## 1. Introduction

All plants coexist with communities of fungi and bacteria in, on, and around them [1,2]. These microbes can colonize aboveground surfaces (the phyllosphere), soil near the roots (the rhizosphere), and the interior of plant tissues (the endosphere) [1,3], and they can significantly contribute to the overall health of the plant [1]. A plant’s microbiota—the collection of all microbes associated with it—can benefit the plant by protecting it from pathogens and herbivores [4,5,6,7], protecting against abiotic stress [8,9,10,11,12] and promoting growth through nutrient acquisition (through nitrogen (N) fixation, phosphate (P) solubilization, siderophore production, etc.) [2,13,14,15,16,17] and phytohormone production [18,19,20]. Beneficial endophytes can activate plant immune responses, resulting in a level of protection from pathogens [21,22,23]. In turn, the host plants affect microbes by changing soil chemistry and secreting signaling compounds [24,25,26,27], exuding energy-rich carbon compounds into the rhizosphere [25], and otherwise providing niches for microbes [28].

An active research area in plant–microbe interactions is determining the extent to which plant genetic variation alters the microbial community [2,29,30,31]. One motivation for this research is the idea of breeding crops for improved microbial associations [29]. At the interspecies level, it has been shown that host species shape the bacterial microbiome throughout several niches [32]. Several studies have shown that intraspecies host genetics significantly affect the microbiome community structure in maize [3,7,33,34,35], rice [36,37], wheat [38,39], and other crops [40,41,42].

Maize has been a model crop for plant genetics for over 100 years [43], due, in large part, to its extensive genetic variation and high economic value (170.7 bushels/acre and USD 9.2 billion in exports alone in 2020) [44,45,46]. Although most commercial maize consists of F1 hybrids [44], most maize microbiome research has been conducted on inbred lines [3,33,47,48], with only a few studies examining the difference between inbred and hybrid maize [34,49].

F1 hybrids show increased vigor and yield relative to their parents [34], an effect called hybrid vigor or heterosis. Heterosis, which is particularly strong in maize, can manifest as increased growth rate, biomass, stress resistance, and yield [50,51].

Recently, it was found that field-grown maize displays heterosis in bacterial rhizosphere communities, as well as fungal communities in the rhizosphere and phyllosphere [34]. In addition, heterosis for germination and root biomass was shown, at least in some instances, to depend upon the local microbial community [49]. In this case, heterosis resulted from inbreds, as well as hybrids, performing under sterile conditions but worse in the presence of microbes. These results indicate that some part of heterosis may be due to the superior ability of hybrids to deal with harmful microbes in the environment.

These previous studies focused on the exterior communities of the plant (the rhizosphere and phyllosphere). In this study, we sought to characterize how inbreeding and heterosis affect both the interior and exterior bacterial communities of maize by looking at the bacteria of the rhizosphere, root endosphere, and stalk endosphere communities in three differently inbred maize groups (inbreds, F1 hybrids, and open pollinated varieties). Our primary goals were to (1) characterize the bacterial communities in each compartment for each group, (2) determine aspects of the community that were consistent across them, (3) determine differences in the communities that could be linked to heterosis, and (4) test the hypothesis that hybrid maize may select superior microbial communities.

## 2. Materials and Methods

### 2.1. Field and Greenhouse Design

Fields were planted in the summer of 2018 (Year 1) and 2019 (Year 2) at different locations within the Iron Horse Research Farm in Watkinsville, Georgia. Plants were grown via standard agronomic practices for the state of Georgia [52]. Our first trial in 2018 consisted of rhizosphere and stalk samples, and there were no root samples for this year. We planted 6 genotypes and sampled 2 plants from each genotype, with a single rhizosphere and stalk sample from the same plant (6 genotypes × 2 tissues × 2 biological reps = 24 samples). Our second trial in 2019 consisted of a single rhizosphere, root, and stalk sample from the same plant. We used 8 genotypes and 4 reps per genotype (8 genotypes × 3 tissues × 4 biological reps = 96 samples). A randomized complete block design was used in the field and greenhouse.

A single greenhouse experiment was carried out in 2019, looking at rhizosphere, root, and stalk samples within the same plant. For each pot, four seeds were planted in a 5-gallon pot with 90% Fafard 3B/Metro-Mix 830 professional growing mix (Sungro Horticulture) and 10% vermiculite. Upon emergence, pots were thinned down to one plant per pot. Three pots per genotype were grown, and pots were arranged in a randomized block design, with each table in the greenhouse consisting of a block containing all 14 genotypes. (14 genotypes × 3 tissues × 3 biological reps = 126 samples).

In total, we had 248 samples across the three experiments. Not every genotype was used in all 3 experiments. Table 1 below shows which genotypes were used in each experiment. Appendix A shows the metadata for each sample, including tissue, experiment, block, number of ASVs, and number of reads.

### 2.2. Sample Collection and Processing

Plants were harvested in a single day to avoid batch effects. A 10 cm section of stalk was cut from the plant 20 cm off the ground using sterilized razor blades and gloves. Plants were dug up around the roots, and roots were removed from the center of the root ball and placed into a clean falcon tube for root and rhizosphere samples.

The outer portions of stalks were removed with a sterile razor blade, and the inner tissues (protected from contamination and external microbes) were cut into 1–3 mm pieces and loaded into a 2 mL conical tube for GenoGrinding (SPEX SamplePrep). Root samples were vortexed at the max setting for 15 s in deionized water to separate the rhizosphere from the root. This wash was then centrifuged at 4500× *g* for 10 min in preparation for DNA extraction. Roots were then thoroughly cleaned with deionized water to remove any residual rhizosphere. Then, 2–3 cm of roots were chopped up with a sterile razor blade and loaded into a 2 mL conical tube for GenoGrinding (SPEX SamplePrep, Metuchen, NJ, USA). 

### 2.3. DNA Extraction and Sequencing

DNA was extracted with a Quick-DNA Fecal/Soil Microbe 96 Kit (Zymo, Irvine, CA, USA) following the manufacturer’s instructions, and 16 s rDNA gene amplification was performed using Earth Microbiome Project 515F [53] and 806R [54] primers with linkers. The sequences are GTGYCAGCMGCCGCGGTAAGT (515F) and GGACTACNVGGGTWTCTAATCC (806R). Peptide nucleic acids (pPNA and mPNA to block plastid and mitochondrial amplification, respectively; PNA Bio) were mixed and diluted to 2.5 μM each for inclusion in the reaction. The first PCR reaction consisted of a 5 μL DNA template, 2 μL of each primer (0.5 μM), 12.5 μL of Hot Start Taq 2X Master Mix (New England Biolabs, Ipswich, MA, USA), 2.5 μL PNA mixture (2.5 μM each), and 1uL of sterile water. The amplification reaction was conducted at 95 °C for 45 s; twenty cycles of 95 °C for 15 s, 78 °C for 10 s, 60 °C for 45 s, 72 °C for 45 s; and finally held at 4 °C. PCR products were purified with AMPure (Beckman Coulter Life sciences, Indianapolis, IN, USA).

Five μL of the first PCR product for each sample was used in the second PCR amplification. The reaction mix consisted of 5 μL of the first PCR product, 5 μL Nextera i5 and i7 Barcode Primers, 12.5 μL 2 × Taq DNA polymerase master mix, and 2.5 μL PNA mix. The second amplification reaction was conducted at 95 °C for 45 s; 25 cycles of 95 °C for 15 s, 78 °C for 10 s, 60 °C for 45 s, 72 °C for 45 s; and finally, 68 °C for 5 min, followed by holding at 4°C. The second PCR products were purified using AMPure beads, and the cleaned products were eluted in 27uL of sterile water and stored at −20 °C until sequencing. Three blanks were used in DNA extraction and library prep. Libraries were sequenced at the Georgia Genomics and Bioinformatics Core on an Illumina MiSeq instrument (Illumina, San Diego, CA, USA) using one paired-end 250 flow cell. The raw data are available in the NCBI Sequence Read Archive under accession PRJNA924784.

### 2.4. Bioinformatics

Sequence processing and quality filtering was completed within the QIIME2 version 2019.1 toolbox [55]. Cutadapt [56] was used to trim primers from raw sequences and filter reads that did not reach a Phred score of 26. FastQC was used to visualize read quality [57]. Paired reads were joined with vsearch in QIIME2 [58]. Deblur [59] was used to truncate reads to 200 bp. The SILVA 132-99-nb classifier [60] was used to assign taxonomy to ASVs. The original dataset contained 15,126 ASVs in 248 samples (excluding blanks). Taxa with no phylum identity were discarded, as well as ASVs found in blanks and ambiguous calls. We also removed taxa related to the host, chloroplasts, and mitochondria, as well as reads that were only present once or twice in a sample. Samples with fewer than 500 reads were removed. ASVs were not agglomerated into OTUs. This left us with 10,922 ASVs. For some analyses (such as core ASVs), we only looked at the 938 taxa that were found in all three experiments.

First, we compared alpha diversity based on genetic background, tissue, and location. We used observed ASVs, as well as Shannon and Simpson indices, from the phyloseq package [61]. Pairwise Wilcox tests and Dunn’s post hoc test were used to test for significance with the FSA package [62]. UniFrac distance matrices were generated in QIIME2 for beta diversity and plotted to visually represent sample diversity. To test which variables had the most significant impact on beta diversity, we generated Bray–Curtis distances in phyloseq; then, a PERMANOVA was performed using vegan [63]. Alpha and beta diversity analyses were performed on data that was rarefied to 500 reads per sample with our seed set to 18. Differential abundance analysis was used to identify ASVs that occurred in all three experiments that differed between inbred and hybrid maize. DESeq2 [64] was used to fit negative binomial models with an alpha value of 0.001. Core ASVs were defined as present in 50% of samples. UpSet plots were created using the UpSetR [65] package. PiCRUST2 [66] was used to predict functional gene pathways from ASVs using the KEGG Orthology database [67]. Raw KO terms were agglomerated to higher functional pathways, and DESeq2 was used to identify pathways that differed between inbred and hybrid maize compartments, with an alpha value of 0.001. Psadd was used to create interactive krona plots of microbiome taxonomy [68,69]. All bioinformatics scripts and pipelines are available at https://github.com/wallacelab/paper-schultz-microbiome-2023, accessed on 20 January 2023.

### 2.5. MiniMaize Inoculation Experiment

Maize lines B73 and Mo17 and their F1 hybrid seeds were sterilized via our previously established method [70]. Seeds were surface-sterilized with sterile water, bleach, and Tween 20, then placed in a hot water bath. They were then allowed to germinate on Hoagland’s agar for seven days to check for contamination. These seeds were then planted in the greenhouse as described above and grown until flowering. Once the silks emerged, stalk sections were sampled from 6–12 inches above the soil line. A razor was used to cut a 10 cm × 10 cm square in the side of mature maize root ball, and roots were removed from the plant to include roots all the way to the center of the pot.

Microbiome extraction was modified from [71,72]. Stalk samples were cored using a sterilized drill tip. Stalk pulp was placed into a 50 mL falcon tube and filled with 40 mLs of MilliQ H_2_O, then shaken 50 times and vortexed at max speed for 10 s. Then, 30 mLs of liquid was decanted into another falcon tube, using the tube cap to exclude large debris. The microbe suspension was centrifuged for 2 min at 3500 RPM to pellet plant debris, 20 mLs which was filtered through a 2 μm Whatman filter. Root samples were placed into a falcon tube without drill tip pulverization, and the same method was used to extract rhizosphere microbiomes.

Sterilized MiniMaize seeds were planted in 2.7 L sterilized pots with autoclave media mixture (same as above). Two sterilized MiniMaize seeds were planted in each pot. Pots were inoculated with 10 mLs of either B73, Mo17, or F1 combined stalk and rhizosphere microbiomes, with 8 pots per treatment. Autoclaved tin foil was placed over the pots for 4 days to ensure no outside microbes were introduced; then, plants were thinned to one plant per pot. Plants were allowed to grow for 5 weeks. Shoots were cut at the soil line and placed in brown paper bags. Roots were gently washed of soil and bagged. Above- and belowground samples were dried and weighed.

## 3. Results

We grew hybrid, inbred, and open pollinated maize lines in two field experiments (2018 and 2019) and one greenhouse experiment. Bacterial microbiomes were extracted from stalks, roots, and rhizospheres of the plants and quantified with QIIME2 and deblur. High-quality reads were retained and classified with the SILVA taxonomy classifier. Low-abundance amplicon sequence variants (ASVs), as well as ASVs associated with mitochondria, chloroplasts, and our blanks, were filtered out. Our final dataset consisted of 241 samples and 10922 ASVs, 938 (9%) of which were present across all three experiments.

Throughout the three experiments, we found that stalk tissue had lower read depth and fewer associated ASVs compared to the rhizosphere and roots. The rhizosphere had the largest number of ASVs, with a majority of these also found in the roots (Figure 1A). Only a fraction of the microbial community found in the rhizosphere and roots can be found in the stalks, although these shared ASVs accounted for the majority (99%) of stalk reads. While the number of ASVs in a phylum appears to accurately represent the relative abundance of reads in the rhizosphere and root, we see differences in the stalk. The relative abundance of ASVs in the rhizosphere and root relative was roughly in line with the number of unique ASVs. However, stalk samples were dominated by proteobacteria reads (38.9% of stalk ASVs but 66.7% of total read depth). Krona plots (nested pie chart distributions) of overall community structures, with comparisons for tissue and genetic background, can be found in the Appendix A.

To investigate which taxa were shared by groups of samples, we plotted intersections of common ASVs collapsed at the genus level (Figure 2). We defined common taxa as genera that were found in at least 50% of samples in a group; a table for all taxa and groups can be found in the Appendix A. Forty-seven genera were shared by at least 50% of samples in inbred and hybrid roots and rhizospheres. A total of 11 genera were shared by inbred roots and rhizospheres but not hybrids, and 8 were shared by hybrids. As a group, the rhizosphere samples contained five genera that were not found in the roots, while the roots comprised three genera not found in the rhizosphere. Inbred and hybrid stalk samples shared four common microbes that were not found in underground compartments. No genera were shared across all samples and genetic backgrounds. All intersections can be found is Appendix A.

Alpha diversity was measured with three common metrics—observed ASVs, Shannon entropy, and Hill’s q1 (exponential of Shannon entropy [73] Kegg Orthology database) on rarefied data (Appendix A)—and compared using Kruskal–Wallis and Dunn’s tests. For all tissue types, we found that field samples had higher alpha diversity than their greenhouse counterparts (*p* < 0.001). Similarly, the root and rhizosphere samples had higher alpha diversity than stalk samples (*p* < 0.001). Post hoc tests show that there were no significant differences in alpha diversity between inbred, hybrid, and open pollinated samples across experiments (Appendix A).

Beta diversity was calculated using the weighted UniFrac metric [74] (Figure 3). Samples were most strongly separated based on tissue type, with rhizosphere, root, and stalks strongly separating from each other. Whether the experiment made a difference depended on the tissue: rhizosphere and root samples were strongly differentiated based on the experiment, and stalk samples were not at all differentiated. Genetic background did not significantly differentiate samples in any compartment. PERMANOVA analysis of weighted UniFrac distances indicated that experiment and tissue type had the most impact on beta diversity (*p* = 0.001 and *p* = 0.001 by Type II ANOVA) (Appendix A). Genetic background (inbred/hybrid/open pollinated) and individual genotype had no significant effect on beta diversity (*p* > 0.05; Appendix A).

To identify the microbes that are most different between samples, we analyzed differentially abundant microbes with DESeq2 (Table 2 and Figure 4), using the 938 ASVs found in all three experiments. Table 2 shows differential comparisons across tissue type, genetic background, and location. We see that location and tissue type have far more differentially abundant microbes than comparisons of genetic background. Most of these were found in the roots and stalks, and many were members of the *Burkholderiaceae* and *Rhizobiaceae* families.

Previous work has shown that metabolic functions provide a better characterization of microbial communities than 16s-based taxonomy [33,75,76]. We used PICRUST2 [66] to predict community functional capacity from the 16s sequencing and DESeq2 to compare differential abundance for comparisons across genetic background, tissue, and location (Table 3). Functional differences were much larger between tissue types and locations than genetic background. When comparing inbred and hybrid tissues, while each compartment had some individually different metabolic functions, only the roots maintained multiple significant differences when individual functional annotations were grouped into metabolic pathways (Table 3). Inbred roots exhibited an increase in predicted gene groups related to metabolism and molecular degradation, while hybrid maize roots exhibited an increase in groups related to carbon transport, electron transfer carriers, and photosynthesis (Appendix A).

Since hybrid maize is generally more fit than inbred maize, we hypothesized that hybrid maize would cultivate a more optimal microbiome. To test this hypothesis, we grew inbred lines B73 and Mo17, as well as their F1 hybrid, to maturity from surface-sterilized seeds germinated in vitro. These plants were grown to flowering in the greenhouse and allowed to recruit microbiomes from non-sterile potting mix. Fast-flowering mini maize [77] was surface-sterilized and germinated in vitro, then inoculated with filtered bacterial microbiomes harvested from stalks and rhizospheres of mature B73 and Mo17, as well as their F1 hybrid, or an autoclaved control. After 28 days, roots and shoots were harvested, dried, and weighed. No significant differences occurred belowground for the separate inoculation groups. Inoculation with microbiomes from B73 (*p* = 0.048) and the F1 hybrid (*p* = 0.037) resulted in a small but significant decrease in shoot biomass relative to the control (Appendix A). No other differences were significant.

## 4. Discussion

Our results show that inbreeding has a small but significant effect on maize microbial communities. However, these effects are much smaller than the effect that environment and tissue compartment have on microbiome makeup and predicted function. One limitation of this study is that due to logistical constraints, not all genotypes were present in all three experiments. These results have several implications for maize–microbe interactions.

First, we showed that root communities were very similar to the rhizosphere soil, while only a fraction of these bacteria can be found in stalk tissue (Figure 1). This may be due to a combination of strong filters as bacteria travel up the plant [78,79] or a larger effect of seed-transmitted microbes in the stalks compared to the roots [78]. It has been shown that endophytes can travel and persist in different tissues in maize [35]. Stalk samples had fewer ASVs overall and lower reads but had a similarly sized common microbiome compared to the roots and rhizosphere (Figure 2). Although shared taxa were significantly different based on tissues, there were no such differences when comparing inbred and hybrid maize.

Similarly, there were no significant differences in alpha diversity measures based on inbreeding, but location and tissue had large effects (Appendix A). These are similar to the results reported by Wagner et al. [34], who also found no significant differences in alpha diversity between inbred and hybrid maize. We found alpha diversity to be higher belowground than aboveground, and we found field microbial communities to have higher alpha diversity than greenhouse communities. A similar pattern held for beta diversity, where tissue and experiment had larger impacts than inbreeding, as shown by both PCoA plots and PERMANOVA (Figure 3). These results add to the emerging collection of data showing that (1) maize tissues have different and distinct microbiomes [3,33,47] and that (2) environment has a larger impact on plant microbiome assembly than intraspecies genetics [3,33,80].

When directly comparing inbred versus hybrid communities, we identified 11 differentially abundant ASVs in the roots, 14 in the stalks, and 2 in the rhizosphere (Figure 4). This represents a very small number of ASVs in this dataset—much smaller than the hundreds of differential ASVs indicated by other characteristics (Table 2). Most of these differences were found in the roots and stalks, indicating that the effect of inbreeding may be most pronounced there compared to the rhizosphere. Perhaps not surprisingly, many of these ASVs belong to groups known to include plant-associated and plant-beneficial bacteria, such as *Rhizobiaceae* and *Burkholderiacea.* Endophytes from both of these families have been shown to have growth promotion potential in maize [15,81,82,83]. We also identified a species of the genus *Pantoea*, which is known to be associated with plants [84,85,86,87], as well as a species of the genus *Sphingomonas*, which has been shown to promote growth and can play a role in phytoremediation [88].

Although predicting metabolic capacity from 16s data is less precise than actual metagenomics data, prior work has shown that metabolomics data support PICRUST2 predictions [89]. It has been found that metabolic functions of the microbiome community may be more important than the taxonomic identity of the individual microbes [75,76,90], and Wallace et al. [33] found predicted metabolic pathways to be more heritable (meaning affected by host genetics) than individual microbes in the maize leaf microbiome. Our data revealed multiple predicted differences in gene functions in maize roots—but not the stalk or rhizosphere—when comparing inbred and hybrid maize. Similar to taxonomic differences (Table 2), the predicted metabolic differences were much smaller when comparing intraspecies inbreeding than when comparing locations and tissues (Table 3). Inbred roots exhibited an increase in predicted gene groups related to acetyl-coA activity, molecular degradation of organic compounds, and increases in plant–microbe signaling, while hybrid maize roots exhibited an increase in groups related to ribosome synthesis, energy, and photosynthetic functions (Appendix A). Taxon contributions to these gene groups (Appendix A) were taxonomically diverse, and most were not found to be differentially abundant in inbred or hybrid roots, highlighting the importance of comparing community function. It has been shown that plant-associated bacterial genomes encode multiple carbohydrate metabolism functions, as well diverse functions related to organic compound metabolism and plant protein mimicry [91]. Many of these functions were found in bacteria cultured from diverse maize seeds [35].

Inbred plants showed increases in acetyl-coA-transferases and dehydrogenases, implying that there may be a difference in anaerobic metabolism, although there were no known acetogens [92] in our differentially abundant ASVs. There were also increases in enzymes related to organic compounds that may be produced by the plant or bacteria. Plants produce exudes and can respond to stress through volatile organic compounds (VOCs), while plant microbiomes can also produce a wide array of VOCs that can impact plant health. These VOCs are incredibly varied and include alcohols, aldehydes, acids, esters, fatty acids, and hydrocarbons (reviewed in [93,94]). We found predicted functions related to esters [93], hydrogen peroxide (plant VOC) ligases [95,96], and phosphonates, which can protect bacteria and make phosphate available to the plant [97,98,99]. These increases may indicate differential communication between the plant and the microbiome, or the microbiome’s reaction to plant stress response. If heterosis is partly microbe-dependent [49], a thorough investigation of these signaling pathways may reveal mechanisms influencing heterosis in maize.

Hybrid plants exhibited an increase in pathways related to ribosome biogenesis (including synthases and GTPase A), energy production, and photosynthesis/carbon metabolism. Several pathways related to energy production and storage were found to be increased in hybrid maize. These included oxidative phosphorylation, heme uptake proteins, and electron transport. Outside of these energy-related pathways, a number of energy and protein pathways related to phototrophic capabilities were differentially abundant in hybrid maize root bacterial communities. These include plastocyanin [100], cytochrome c subunit [100], and photoreaction center m [101,102], all of which are related to photosynthetic energy production in photoreactive bacteria [102]. Cyanobacteria ASVs found within these maize microbiomes (Figure 1) had little effect on the differences in functional gene prediction. Differences in photosynthetic protein and photosynthesis gene groups were due to a wide range of phyla, including Proteobacteria, Chloroflexi, Fimicutes, and Acidobacteria (Appendix A), all of which have been shown to have phototrophic capabilities [103]. These taxa were not identified as differentially abundant (Figure 4). This analysis indicates that hybrid plants may select for microbial communities that have increased energy production or phototrophic potential. Analysis of actual metagenomic data instead of 16s-based predictions would allow us to parse apart real metabolic differences between the two communities, whereas this analysis focused on predicted potential of the communities.

When we inoculated mini maize with the filtered bacterial microbiome from B73, Mo17, and their F1 hybrid, we saw minor differences in biomass aboveground but not belowground (Appendix A). Specifically, we saw B73 and F1 inoculates decreased plant biomass compared to control inoculated plants, while Mo17 had no significant effect. We hypothesized that hybrid maize may harbor more beneficial bacteria, but our results do not support this conclusion. In a previous study, Kaeppler et al. [104] tested the mycorrhizal responsiveness of 28 inbred maize lines. It was found that B73 and Mo17 had the largest differences in response to mycorrhizae. In our study, we used bacterial filtrates, indicating that these two lines may consistently respond differently to microbial inoculation.

Our results are somewhat similar to those reported by Wagner et al., who showed significant differences in inbred and hybrid maize microbial communities in the field [34] and that at least some effects of heterosis may be due to the superior ability of hybrids to deal with harmful microbes [48].

Our primary goals were to (1) characterize the bacterial communities in each compartment for each group, (2) determine aspects of the community that were consistent across groups, (3) determine differences in the communities that could be linked to heterosis, and (4) test the hypothesis that hybrid maize may select superior microbial communities.

We found that (1) bacterial communities in the roots and rhizospheres were very similar to each other and that stalk communities only have a small portion of these bacteria; (2) common bacteria were present across compartments, regardless of inbreeding; (3) intraspecies heterosis played a much smaller role in microbial community diversity, composition, and function than tissue compartment or location; and (4) in a small greenhouse experiment, hybrid microbiomes failed to benefit inoculated mini maize.

## 5. Conclusions

Modern maize breeding is based on heterosis, and with many new biologicals coming to market, it is paramount to understand how these microbes interact with intraspecies plant genetics. The literature has shown that host species play a large role in shaping niche microbiome, and our data show that inbreeding has small and significant effects on taxa and function in maize microbial communities. However, these effects pale in comparison to the effects of environment and tissue type on community composition. With the environment playing such a large role in shaping bacterial communities, investigating the extent to which maize can recruit specific taxa/functions would shed light on the potential to use the plant itself to alter its microbiome. Further work is needed to examine how maize genetic diversity and the environment shape community function. Understanding how the microbiome interacts with maize genetic diversity will allow breeders and scientists to make better use of microbial communities for more sustainable crop production.

## Figures and Tables

**Figure 1 microorganisms-11-00879-f001:**
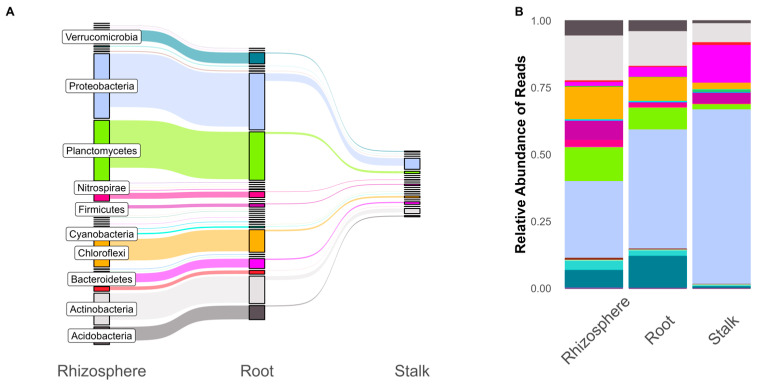
Shared ASVs across plant tissues (**A**) compared to relative abundance of reads in plant tissues (**B**) colored by phylum.

**Figure 2 microorganisms-11-00879-f002:**
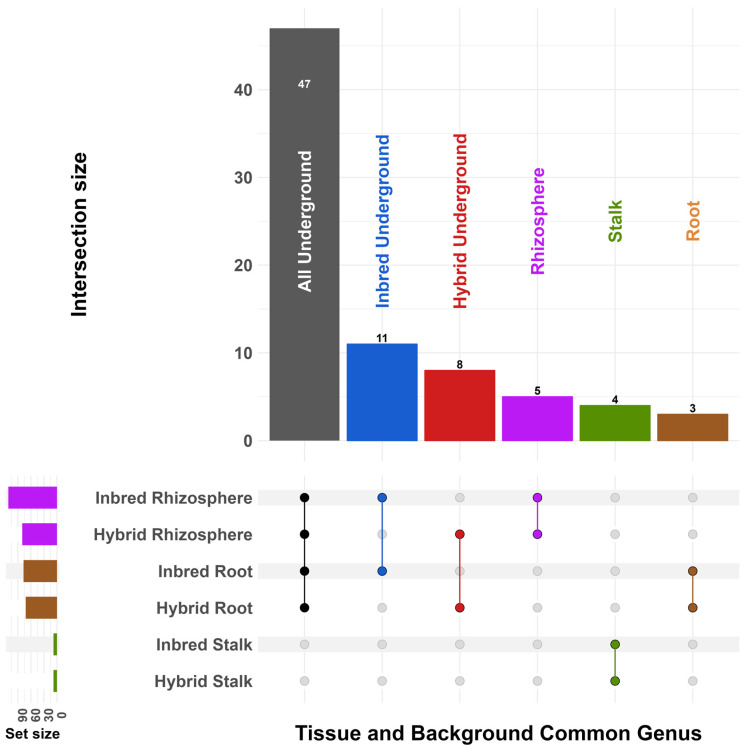
UpSet plot [65] showing intersections of common genera (those found in >50% of samples) based on genetic background and tissue. Intersections with zero counts are not shown. Open pollinated lines share genera in some but not all of these intersections.

**Figure 3 microorganisms-11-00879-f003:**
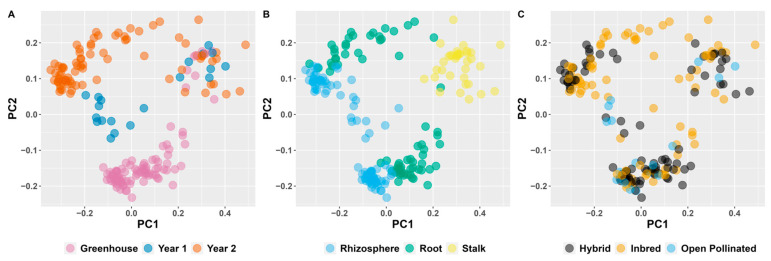
Weighted UniFrac diversity principal coordinates colorized by experiment (**A**), tissue type (**B**), and genetic background (**C**).

**Figure 4 microorganisms-11-00879-f004:**
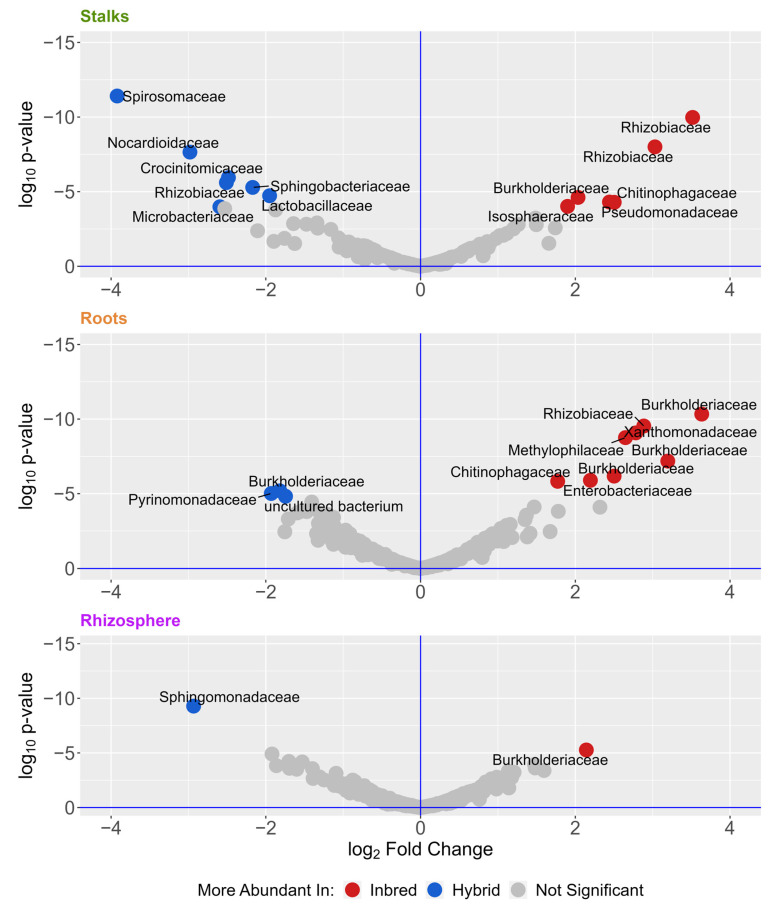
Volcano plots of differentially abundant ASVs. ASVs were more abundant in inbred (blue) or hybrid (red) determined by DESeq2 with an alpha of 0.001. Dots represent individual ASVs, which are labeled according to their taxonomic family. The full list of differentially abundant ASVs is presented in Appendix A.

**Table 1 microorganisms-11-00879-t001:** Maize genotypes used in the three experiments.

Maize Genotype	GRIN Accession	Genetic Group	Experiment
CML247	PI 692141	Inbred	GH
Mo17xPh207		Hybrid	GH
Mo17	PI 558532	Inbred	GH, Year 2
Reid Yellow Dent	PI 222613	Open Pollinated	GH, Year 1
Ph207	PI 601005	Inbred	GH, Year 1, Year 2
Ph207xB73		Hybrid	GH, Year 1
B73	PI 550473	Inbred	GH, Year 1, Year 2
Oh43	PI 690332	Inbred	GH
B73xCML247		Hybrid	GH
B73xOh43		Hybrid	GH
Mo17xB73		Hybrid	GH
B73xMo17	Ames 19097	Hybrid	GH, Year 2
Hopi_blue	NSL 165817	Open Pollinated	GH
B73xPh207		Hybrid	GH, Year 1
DKC70-27		Hybrid	Year 2
903VIP		Hybrid	Year 2
CML322	PI 690321	Inbred	Year 2
HP301	PI 587131	Inbred	Year 2
Bloody Butcher	Ames 32345	Open Pollinated	Year 1

**Table 2 microorganisms-11-00879-t002:** Table of differentially abundant taxa. DESeq2 was used to compare genetic background, location, and tissue type. Tissue type and location had a much larger impact on the number of differentially abundant ASVs than genetic background.

Tissue	Comparison	Number of ASVs
Compared genetic background		
All	Inbred vs. Hybrid	61
All	Inbred vs. Open Pollinated	76
All	Hybrid vs. Open Pollinated	20
Rhizos	Inbred vs. Hybrid	2
Rhizos	Inbred vs. Open Pollinated	8
Rhizos	Hybrid vs. Open Pollinated	5
Roots	Inbred vs. Hybrid	11
Roots	Inbred vs. Open Pollinated	6
Roots	Hybrid vs. Open Pollinated	2
Stalks	Inbred vs. Hybrid	14
Stalks	Inbred vs. Open Pollinated	7
Stalks	Hybrid vs. Open Pollinated	6
Compared locations		
All	Field vs. Greenhouse	504
Rhizos	Field vs. Greenhouse	192
Roots	Field vs. Greenhouse	182
Stalks	Field vs. Greenhouse	33
Compared tissues		
-	Stalks vs. Rhizos	512
-	Stalks vs. Roots	371
-	Roots vs. Rhizos	274

**Table 3 microorganisms-11-00879-t003:** Differentially abundant PICRUST-predicted genomic functions. Agglomerated pathways were grouped based on KEGG pathways before differential abundance was determined (alpha of 0.001).

Tissue	Comparison	Agglomerated Pathways	Raw Annotations
Compared genetic background			
All	Inbred vs. Hybrid	1	261
All	Inbred vs. OP	0	846
All	Hybrid vs. OP	0	192
Stalks	Inbred vs. Hybrid	2	26
Stalks	Inbred vs. OP	1	334
Stalks	Hybrid vs. OP	0	139
Rhizosphere	Inbred vs. Hybrid	0	28
Rhizosphere	Inbred vs. OP	0	29
Rhizosphere	Hybrid vs. OP	0	67
Root	Inbred vs. Hybrid	34	638
Root	Inbred vs. OP	0	170
Root	Hybrid vs. OP	0	21
Compared locations			
All	Greenhouse vs. Field	48	2994
Stalks	Greenhouse vs. Field	13	724
Root	Greenhouse vs. Field	108	4489
Rhizosphere	Greenhouse vs. Field	176	4966
Compared tissues			
-	Stalks vs. Roots	139	5261
-	Stalks vs. Rhizosphere	165	5901
-	Rhizosphere vs. Root	71	3587

## Data Availability

All bioinformatics scripts and pipelines are available at https://github.com/wallacelab/paper-schultz-microbiome-2023, accessed on 20 January 2023. The raw sequence data are available in the NCBI Sequence Read Archive under accession PRJNA924784.

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
