# Peer review of "Effects of Inbreeding on Microbial Community Diversity of Zea mays"

_microorganisms, 2023, doi:10.3390/microorganisms11040879_

Round 1

Reviewer 1 Report

Schultz et al. designed an interesting experiment and acquired abundant data for the purpose to illustrate the effects of inbreeding on microbial community diversity of Zea mays. And based on the analyses to the microbiomes of maize stalk, root and rhizosphere, the authors come to a conclusion that the non-genetic effects are generally the largest influences on the maize microbiome. I have a question why authors could make such a conclusion only based on the data of intra-species taxonomical plants. Plant taxonomical level of species or above may observe the effects on shaping their microbiomes. As a result, to make this conclusion should add a limitation. May suggestion to the manuscript is to make full use of the data obtained, investigating the exchanges of microbial ASVs among maize microbiomes. In addition, the writings and presentation of the manuscript are far from satisfactory: results presented with many discussions; figure legends included many conclusions; and in discussion section duplicated lots of the results. To change the manuscript as other styles instead of a research article may be better.

Author Response

Reviewer 1

Schultz et al. designed an interesting experiment and acquired abundant data for the purpose to illustrate the effects of inbreeding on microbial community diversity of Zea mays. And based on the analyses to the microbiomes of maize stalk, root and rhizosphere, the authors come to a conclusion that the non-genetic effects are generally the largest influences on the maize microbiome. I have a question why authors could make such a conclusion only based on the data of intra-species taxonomical plants. Plant taxonomical level of species or above may observe the effects on shaping their microbiomes. As a result, to make this conclusion should add a limitation.

Reply:

Throughout the paper we qualified the effects of genetic background and heterosis as intra-species. We also included references to the magnitude that host species plays on shaping the microbiome in both the introduction and conclusion. 

May suggestion to the manuscript is to make full use of the data obtained, investigating the exchanges of microbial ASVs among maize microbiomes.

Reply:

Unfortunately, with our data only containing a single time point, it is impossible to identify bacteria that move from one tissue to another. Without multiple time points, or tagged bacteria in a controlled experiment, it is impossible to accurately identify the origin or movement of these taxa.

In addition, the writings and presentation of the manuscript are far from satisfactory: results presented with many discussions; figure legends included many conclusions; and in discussion section duplicated lots of the results. To change the manuscript as other styles instead of a research article may be better.

Reply:

While we appreciate Reviewer 1’s feedback on manuscript writing style, no other reviewer asked for changes in manuscript structure. With that in mind the following changes were made: All figure legends have been edited to remove conclusions and interpretations. We ensured the results section informed readers of the conclusions of figures, tables, and analyses, without going into a deeper discussion of what these results mean. While some results are reiterated in the discussion section, we firmly believe this allows readers to better comprehend results, and ensures readers do not have to jump back and forth between results and discussion.

Reviewer 2 Report

The authors of this study reveal intriguing discoveries about the effect of heterosis on the maize microbiome. The findings of this study are interesting and will undoubtedly lead to better understanding of the role of plant genotype on modifying the structure of plant-associated microbial communities. The authors have also explained the methodology properly and in a reproducible manner. In my opinion. the manuscript can be accepted for publication.

Author Response

There were no revisions requested. Thank you. 

Reviewer 3 Report

The manuscript written by Schultz et al. which provides insights into the Effects of Inbreeding on Microbial Community Diversity of Zea mays is novel and scientific. The presentation is easy to follow and informative about the microbes associated with maize. Nevertheless, some corrections are required.

The English language of the MS needs minor checks.

Minor comments.

L33: Write N and P in full at the first mentioning

L71: Is it a Table?

L119, 122: capitalize QIIME

Author Response

The manuscript written by Schultz et al. which provides insights into the Effects of Inbreeding on Microbial Community Diversity of Zea mays is novel and scientific. The presentation is easy to follow and informative about the microbes associated with maize. Nevertheless, some corrections are required.

The English language of the MS needs minor checks.

            Fixed typos and extra periods/spaces.

Minor comments.

L33: Write N and P in full at the first mentioning

            Done

L71: Is it a Table?

            Yes. Caption was added and other tables were renumbered throughout the manuscript.

L119, 122: capitalize QIIME

              Done
Thank you for your feedback. 

Reviewer 4 Report

This study starts from a strong set of research questions and follows a very straightforward and easy-to-follow approach to finding the answers to these questions.

Still, in the present form of the manuscript, I find myself unable to really understand the significance of the presented results and the soundness of the methods as the Material and Methods section gives no sense of the dimension of the experiments nor the number of involved samples:
- It is stated that for each pot there are 4 seeds, then thinned down to 1, and three randomized blocks (per genotype, I assume? not stated), but how many pots per genotype?
- How many plants were considered in field conditions? Was there randomized design in field as well? Not stated.
- How many samples of roots, rhizosphere and stalks were gathered per plant? How many per genotype?
- The authors state that they used 252 samples for their sequencing. While this number is indeed quite high, it must also be said that their experimental design is extremely extensive with several variables. By my simple math, they have 19 genotypes, cultivated in 3 conditions, and 3 starting matrixes (stalk, root, rhizosphere): this means 171 types of samples. Theoretically, this brings me to question if every sample type had at least 2 indipendent sequences or not, as 252 is not enough to have 171 samples ran twice, even less in 3 independent replicates.

This question of sample numbers brings me to my greatest concern regarding analysis: the authors say that they removed ASV that were present in less than 5% of samples. 5% of 252 is 12.6 samples. Then the authors state that there is no influence of genotype and genetic background on the analysis... but do they have at least 13 samples for each genotype? If they do not, it is evident that every genotype-specific AVS would be filtered out before analysis of diversity, thus risking to discard any actual diversity that comes from individual genotpye. Also, considering the strong differences in root, rhizosphere, and stalk communities, it would need that number of samples per genotype PER compartment to be a valid statement.

In conclusion, the authors must provide a clear table of samples, showing the number of reads and AVS obtained for each of them, as the soundness of the conclusions is impossible to understand without these data. They are advised to reconsider their cutoff parameters for "rare" AVS at 5% to a lower number that better reflects the huge number of variables in their experimental setup.

I can give a more thorough evaluation of the conclusions and relevance of the work once these doubts on the methods and design of the expeirments have been cleared.

As minor comments:

- I'd advise the authors to change the definition of "Field 1" and "Field 2" as they imply that the experiments were carried out in parallel in two different fields, instead the variable is the Year, not the Field.

- Regarding the literature of maize-associated microbiota, I'd suggest adding these two references:
https://doi.org/10.1371/journal.pone.0020396 
https://doi.org/10.3390/microorganisms9112388

Author Response

Major Revisions:

- Better description of methodology

Reply:

We better described the number of genotypes and samples in each experiment in the methodology. We emphasized that tissue samples were taken from the same plant for a single biological rep in a single block. We included columns showing the number of ASVs and reads for each sample in the supplementary metadata. We added: One limitation of this study is that due to logistical constraints, not all genotypes were present in all three experiments.

- Cutoff parameters for 5%

Reply:

We thank the reviewer for their concern over the 5% filtering step. We substituted this step with a filtering step removing reads that only appeared 1 or 2 times. This drastically increased the number of rare taxa, and we reran all analyses with this new dataset. Overall this changed some figures and results (magnitude and quantity), while not significantly changing the overall results of the manuscript. To summarize the effects of this new threshold in the revised manuscript:

Figure 1 – Sankey Flow and Relative Abundance: Increased the number of ASVs in both subplots. These are descriptive changes.

Figure 2 – Upset Plot: Increased the number of core taxa by 5, and stalk samples now have more shared core taxa than roots.

Figure S1 – Alpha Diversity: The increase of rare taxa increased overall richness measures, but did not significantly change conclusions from the original draft.

Figure 3 – Beta Diversity: There were no significant changes in the Bray Curtis PERMANOVA compared to the original draft. Weighted Unifrac figures still show more distinct differences in (a) experiment and (b) tissue type, rather than (c) genetic background

Figure 4 – Differentially Abundant ASVs: There was an increase in differentially abundant ASVs, most noticeably in the stalks.

Figure S2 – Predicted Functional Genes: Most notably, differences in predicted functional genetics were found in the stalk (2 pathways) with the new filtering threshold (there were 0 in the original draft). The number of pathways found to be differentially abundant in hybrid and inbred roots actually decreased, however the predicted gene groups stayed very similar. Although the magnitude has changed, the discussion and potential impact remains the same as the original draft. Table 3 had changes in the values, but this did not change overall trends. Interestingly there were few changes when comparing tissues across locations compared to the original draft.

Figure S3 – Greenhouse Microbiome Inoculation: this experiment on biomass does not depend on the microbiome analysis.

To summarize: Changing the filtering step of ASVs found in at least 5% of the samples to removing singleton and double reads greatly increased the amount of reads and ASVs available for this analysis. Overall, these changes slightly changed the magnitude and quantity of differences, but did not significantly alter the paper’s overall results. Most importantly, greater changes were found in stalk samples compared to the original draft. Tables have been updated, new figures have been substituted, new supplementary files were created, methodology, results, and discussion have been changed to reflect these revisions. The reviewer’s observation of this filtering step were absolutely accurate, and all analyses were rerun to reflect the new filtering step.

As minor comments:

- I'd advise the authors to change the definition of "Field 1" and "Field 2" as they imply that the experiments were carried out in parallel in two different fields, instead the variable is the Year, not the Field.

Reply: The two field experiments were carried out on both different fields and different years. In the manuscript as well as Figure 3, these have been changed to Year 1 and Year 2

- Regarding the literature of maize-associated microbiota, I'd suggest adding these two references:
https://doi.org/10.1371/journal.pone.0020396 
https://doi.org/10.3390/microorganisms9112388

Reply:

We included both of these references in the introduction and the discussion. They were relevant to the background and overall context of our experiments. 

Round 2

Reviewer 4 Report

The authors amended the manuscript as requested, both explaining in much finer detail the composition and number of the samples and changing the thresholds to describe the microbiota of their highly diverse sample pool.

The limitations of the work regarding replicates and the presence of all genotypes are - of course - still in the manuscript but they are addressed by the authors, allowing the reader to evaluate the results appropriately.

A single extra comment I'd make: I would strongly suggest to use a color-blind-friendly color palette for the figures.
Figure 1 in particular makes use of a gradient palette with very little contrast that makes it hard for me - and I'm not color-blind - to tell the difference between colors in the (b) part of it, and would therefore be quite impossible to read for color-blind people. The same could apply to Figure 3.

Author Response

Figures 1 and 3 were edited to be more distinct and color blind friendly. Figure 3 used a tradition color-blind friendly palette. Figure 1 had upwards of 30 different colors. We used a bright and distinct color palette in figure 1, and used the pal.safe() function in R to ensure the colors were distinguishable to different visual impairments.